# Plant recruitment six years after the Samarco's tailings-dam disaster: Impacts on species richness and plant growth

André Araújo da Paz[1,2]*, Diego Shimizu-Aleixo[1], Astrid de Oliveira-Wittmann[2], Florian Wittmann[2]*, Cleberson Ribeiro[1], Ricardo Ildefonso de Campos[1]

1 Graduate Program in Ecology, Federal University of Viçosa (UFV), Viçosa, Minas Gerais, Brazil,
2 Department of Wetland Ecology, Institute of Geography and Geoecology, Karlsruhe Institute of Technology (KIT), Rastatt, Baden-Württemberg, Germany

* florian.wittmann@kit.edu (FW); andrearaujodapaz@gmail.com (AAP)

## Abstract

One of the greatest tragedies in Brazilian mining history occurred in November 2015 in Mariana, Minas Gerais state, when a dam from the mining company Samarco was breached. Millions of mine tailings from this upstream embankment were dumped over the Doce River basin, impacting an area of approximately 1469 ha of riparian vegetation. Our objective was to experimentally investigate whether plant recruitment and establishment are impaired in areas affected by tailings six years after the deposition. To achieve this goal, in 2021 we compared soil chemical properties between affected and unaffected areas, performed a soil seed bank experiment in controlled conditions, and conducted a greenhouse growth experiment using the two most abundant plant species. Affected soils presented lower fertility and organic matter content. At the same time, the mean abundance and richness of emerging plants did not differ between soils. Still, affected areas exhibited approximately 35% lower accumulated species richness (gamma diversity) than unaffected ones. The three most abundant species in both areas represented 34% of the individuals, being *Marsypianthes chamaedrys* (Vahl) Kuntze, *Ludwigia octovalvis* (Jacq.) P.H. Raven and *Ageratum conyzoides* L. In the growth experiment, plants growing in affected soils presented reduced height and stem diameter increment (*L. octovalvis*) or allocated fewer resources to root production than aerial parts (*M. chamaedrys)*, potentially in response to soil infertility and density. Even after six years, our results showed that tailings-affected areas continue to experience negative impacts on plant recruitment, highlighting its adverse effects on ecosystem functions and services.

## Introduction

Large-scale mining generates several environmental impacts by altering the landscape before, during and after the ore extraction [1]. In addition, the establishment

**Data availability statement:** All the data was made available at the Zenodo repository, by the following identifiers: DOI: 10.5281/zenodo.15579256 https://zenodo.org/records/15579256?token=eyJhbGciOiJI-UzUxMiJ9.eyJpZCI6ImRlMWFlNTc0LTlmY-mItNDIyNS1hNjRmLWI3YzBhZTRkMjE0M-SIsImRhdGEiOnt9LCJyYW5kb20iOiIzNzdhMz-JhYzQ5MDcxOWMxZGEzZmM4ZDAyYjAzM-2VmZiJ9.9F5QYt5M1hkutgkan_nYHJKWiUh-F2QYESHPLSiI89mD0EY1GdsX1iq0rfLu9_qyvSg2aSwj42drpdvoaVB1HvQ

**Funding:** The authors declare that this research was funded by FAPEMIG/RENOVA*. This agencies were responsible for grants for scientifical initiation (DA) . Also, this study was supported by one PhD grant (AP) from CAPES (Brazil) and DAAD (Germany). The funders had no role in study design, data collection and analysis, decision to publish, or preparation of the manuscript. * This project/product was funded by the Renova Foundation, by imposition of the Transaction and Conduct Adjustment Term – TTAC, signed for the recovery, mitigation and compensation of the socioeconomic and socio-environmental impacts of the Fundão dam collapse, in Mariana, Minas Gerais. The authors declare no financial or non-financial competing interests related to this study. https://www.fundacaorenova.org/https://www.fapemig.br/pt/https://www.gov.br/capes/pt-brhttps://www.daad.de/de/

**Competing interests:** The authors have declared that no competing interests exist.

and operation of mining activities can cause true disasters [2]. A tragic example occurred in November 2015 in Mariana, Minas Gerais state, southeast Brazil [3] when the Fundão dam (Samarco's Company), containing iron ore tailings, burst. A total of 39.2 million $m^3$ of slurry were released into the Doce River basin [4]. The disaster impacted an estimated area of approximately 1469 ha of natural vegetation [5], burying or carrying away the riparian plants and their diaspores in the soil.

After the passage and deposition of this large amount of tailings, the vegetation recovery where likely depended on three main factors: i) the existence of a soil seed bank [6]; ii) the arrival of propagules through dispersal processes [7] and/or iii) restoration practices such as the addition of seeds [8,9]. The following months after the disaster, there were some reclamation procedures such as the sowing of seeds, tree planting, and soil amendment [10,11]. However, the presence of seeds alone does not guarantee the establishment of seedlings, and the recruitment and assembly of the plant community also depend on the survival of seedlings [12]. These stages are susceptible to environmental factors [13] and might be strongly affected by the permanence of the slurry.

The sludge passage directly affected the soil by changing its chemical and physical properties [14]. For example, the deposited tailings differ from the predominant soil in the region by its higher sand content [15], higher density, and lower macroporosity and clay/silt content [16]. Furthermore, in the affected banks of the Doce river tributary, Gualaxo do Norte, higher concentrations of bismuth (Bi), cerium (Ce), chromium (Cr), iron (Fe), manganese (Mn), phosphorus (P), and lead (Pb) were found compared to unaffected areas [16]. While some of these elements are essential for plants, others can be toxic in excessive amounts. For instance, the total contents of Cr ($235.0 \pm 69.5$ mg/kg) and Pb ($52.7 \pm 15.9$ mg/kg) were higher than the soil quality guideline values, which are 75 and 19.5 mg/kg, respectively [16].

The changes in soil properties in the affected areas could alter the restoration process's success in the initial years after the disaster. Considering that various natural and anthropogenic factors serve as environmental filters [17], affected soils can impact plant recruitment by reducing emergence, establishment, and growth in contaminated soils. This can occur by decreasing water absorption, gas exchange, radicle fixation, and substrate foraging [18]. Also, plant germination and establishment can be reduced once the penetration of roots is affected by the soil compaction promoted by the tailings [15]. It has been found that soils contaminated by iron ore tailings might be responsible for lower plant biomass production [19], change in leaves properties [20] and accumulation of potentially toxic elements [21]. It is, therefore, reasonable to expect that a smaller number of plants and fewer species will be able to germinate and grow in affected areas, with practical implications for restoration.

The present study aims to experimentally test if plant recruitment is impaired in areas affected by the passage of the tailings six years after the Samarco dam breach. We analysed soil characteristics as the primary ecological mechanism influencing plant recruitment, growth, and soil seed bank composition. More specifically, we performed experiments in controlled growth conditions to test the following hypothesis:

there is a reduction in i) abundance; ii) richness of emerging seedlings, and iii) plant growth, when comparing affected to unaffected areas.

## Materials and methods

### Study area

We sampled the Upper Doce River basin in Minas Gerais in southeast Brazil (Fig 1). Semideciduous seasonal forests characterize the areas [22] mainly on oxisols in an agricultural matrix of different crops and artificial cattle grazing grasslands, with a long history of degradation [23]. The climate according to Köppen's classification is Cwa and Cwb, humid subtropical with drought in winter and summer from hot to temperate, with annual rainfall of 1200 mm [24]. Map was created using software QGIS 3.34.14 [25].

We sampled soils in three regions between the Fundão dam and the Risoleta Neves HPP (Fig 1). The choice of sampling areas considered the different environments composing the slurry pathway, from the Fundão dam to the Risoleta Neves HPP. This limit was established since the slurry had accumulated in this reservoir and lost power after this point. Thus, we selected areas where the riparian vegetation was directly impacted, showing visible signs of the slurry on the trunks of nearby trees. The areas near Fundão dam have a history of mining activity, while the other downstream areas are primarily used for agriculture and livestock [4]. Using similar geomorphology and vegetation types, we compared the floodplains of affected (A) rivers and unaffected (N) nearby tributaries (Fig 2). In those areas, we sampled soils for three purposes: i) to analyze soil seed bank, ii) for chemical analysis, and iii) for plant growth experiments.

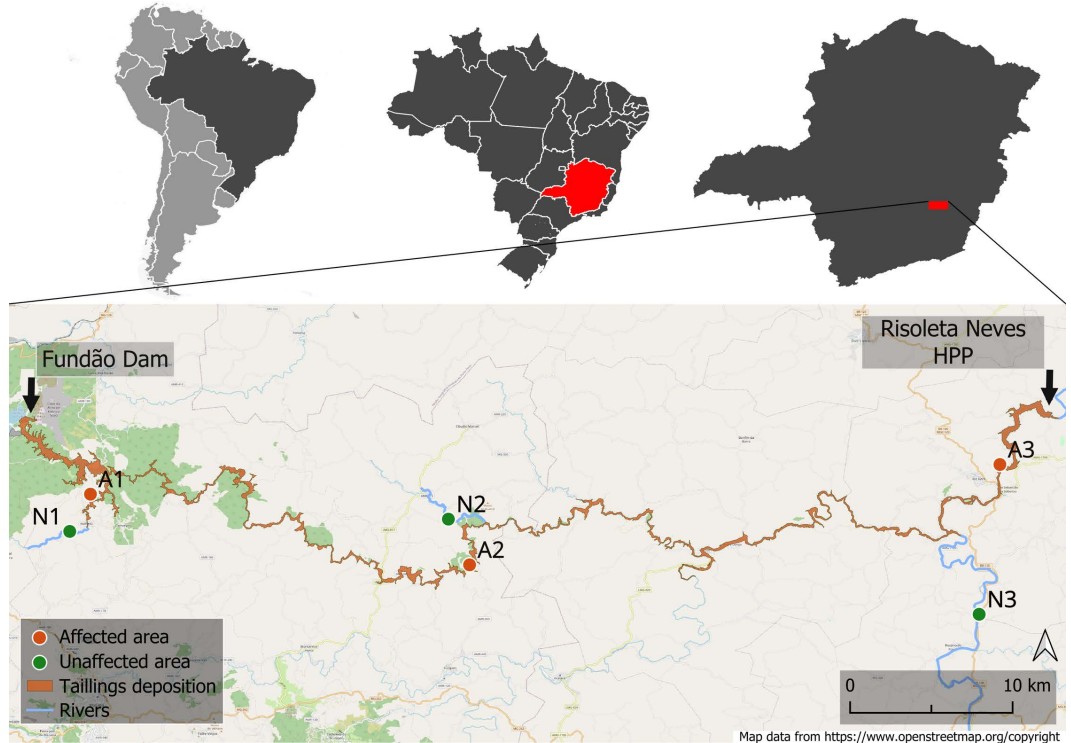

**Fig 1. Location of the areas sampled in the three regions of the study.** Above, from left to right: South America, Brazil, and the state of Minas Gerais. Bellow: Sampling carried out in the regions: 1) downstream of the Fundão dam, 2) downstream of Paracatu de Baixo and 3) upstream of the Risoleta Neves Hydroelectric Power Plant. The tailings deposition zone is highlighted along the sampled region (shape file "PG23_Area_Afetada_Lama" provided by Fundação RENOVA). Map created using QGIS 3.34.14. Map data from openstreetmap.org/copyright.

## Soil collection for seed bank experiment

We sampled soils for the seed bank experiment in three regions between the Fundão dam and the Risoleta Neves HPP (Fig 1). The affected areas and their characteristics are as follows. A1) Gualaxo do Norte River, close to Fundão Dam, with mostly forested floodplains (20º 15' 13"S, 43 º 25' 16"W); A2) Gualaxo do Norte River, near Paracatu de Baixo, with fragments of Atlantic forest mixed with pasture (20º 17' 43"S, 43º 11' 51"W); and A3) Doce River, upstream of the HPP, in a mainly farming and livestock landscape with less forest fragments (20º 14' 10"S, 42º 53' 05"W). Additionally, in the unaffected areas, we sampled the riparian zone of non-affected tributaries, with characteristics similar to those of the affected areas. N1) Gualaxo do Norte River (20º 16' 32"S, 43 º 26' 00"W); N2) Bucão stream (20º 16' 06"S, 43º 12' 36"W); and N3: Piranga River (20º 19' 28"S, 42º 53' 49"W) (Fig 1).

For the seed bank experiment, we sampled in two different seasons to perform a broader and more efficient sampling as plant phenologies vary in time [26]. We sampled during the peaks of the rainy season (Dec/2020) and the dry season (Jul/2021) in the same areas. In each studied area (Fig 1) and each season, we selected 5 points at a minimum distance of 300 m from each other, 15 points in affected areas and 15 in unaffected areas, totaling 30 samples per season. At each point, we collected the topsoil and the litterfall using a metal jig (25 × 25 cm) at 5 cm soil depth. The soil samples were kept in plastic bags at 15°C and were transferred to the laboratory. In the dry season (Jul/2021), four samples from unaffected areas were lost due to technical problems. The assessed areas were private, and the landowners were previously contacted for research approval. Since the sites are out of any conservation unit, no licence was required by Brazilian government.

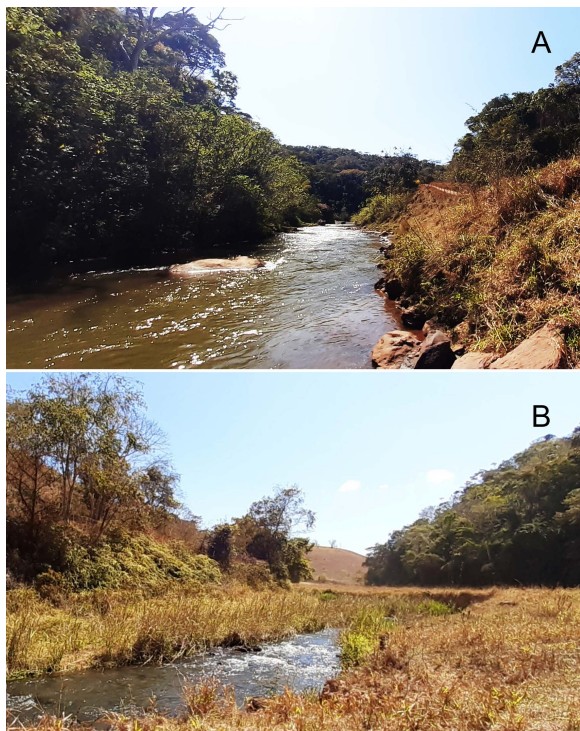

**Fig 2. Riverscapes of areas in the upper Doce River basin during the dry season.** Areas (A) affected by the Fundão dam disaster and (B) unaffected tributaries.

## Seedling emergence experiment in controlled growth conditions

We transferred the sampled soils to plastic trays (0.25 m × 0.5 m × 0.1 m high) and kept them in a germination room, with controlled conditions at 25 (± 2)°C, 70% humidity, with a photoperiod of 12 h. The soil samples were watered daily. To record plant recruitment from the soil seed bank, we used the seedling emergence method, adapted from [27]. In this way, we counted all plant individuals out of each plastic tray for six months (Fig 3). Once a month we removed all the plants from the plastic trays to count and morphotype them. We considered "emerging plants" the ones presenting a complete formation of leaves from at least three nodes. For identification purposes, at the first occurrence of each plant species, we transplanted the specimen into plastic pots filled with greenhouse soil for growing and flowering. Finally, we only considered dicotyledon plants.

The species were identified by observing the vegetative characteristics and, when possible, reproductive characteristics of the plants, with the help of identification manuals [28–31] and local experts. Species names and authorities were standardized using the Taxonomic Name Resolution Service provided by the Missouri Botanical Garden [32]. To verify whether the species are native or not, thinking about their usage in future restoration projects, we used the Flora e Funga do Brasil website [33] to classify them as native or naturalized [34]. The same nomenclature of morphotypes was used in the different seasons (e.g., Asteraceae 1 is the same morphospecies in both seasons). Some individuals could not be identified at the species level.

## Soil collection for chemical analysis

We performed another soil collection for chemical and nutritional analysis and plant growth experiments. Those soil samples were taken from one region around the affected HPP's lake (20º 14' 10"S, 42º 53' 05"W), and in the banks of the Piranga River, the closest unaffected tributary near the reservoir (20º 19' 28"S, 42º 53' 49"W) (Fig 1, areas A3 and N3, respectively). In each of those affected and unaffected areas, we sampled three transects located at least 300m apart from each other. Within each 10 m long transect, we collected three sampling points, each consisting of three subsamples of homogenized soil. In total, we sampled nine points in each of those affected and unaffected areas.

We kept the soil samples in plastic bags at 15°C and transported them to the laboratory, where they were dried and later weighed, detorted, and sieved. The soil was divided to obtain subsamples for chemical analysis at the Soil Physics Laboratory of the Federal University of Viçosa – UFV [35]. In the chemical analysis, we measured pH, exchangeable acidity, sum of bases, cation exchange capacity, base saturation, and organic matter content. We also measured the

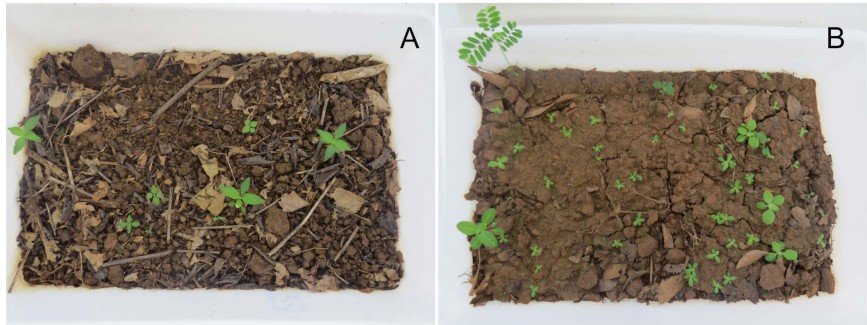

**Fig 3. Substrate samples from riverine banks spread in plastic trays for seedling emergence counting.** (A) Substrate from affected areas and (B) soil from unaffected tributaries.

bioavailability of macronutrients such as phosphorus (P), potassium (K), sodium (Na), calcium (Ca), magnesium (Mg), and the bioavailability of potentially toxic elements such as iron (Fe), manganese (Mn), copper (Cu), zinc (Zn), nickel (Ni), chromium (Cr), cadmium (Cd) and lead (Pb). The extractor used for Ca, Mg and Al was KCl 1 mol/L, and for the other elements, it was Mehlich-1.

## Plant growth experiment

To assess the impact of the soil change following the dam rupture on plant growth, we chose the two most common plant species from the seed bank: *Ludwigia octovalvis* (Jacq.) P.H.Raven (Onagraceae) and *Marsypianthes chamaedrys* (Vahl) Kuntze (Lamiaceae). Both species are annual herbs, *L. octovalvis* being an amphibious plant, between 0.3 to 2.5 m tall [36] and *M. chamaedrys* an aromatic plant with 0.1–1.5 m tall [37]. We selected twenty individuals from the seedling emergence experiment, 10 of each species. The individuals had at least three nodes with fully developed leaves and approximate height. According to their origin in the previous experiment, the individuals were planted in two types of substrates: plants originating from affected areas were transferred to soil from affected areas (A), and plants from unaffected areas were transferred to soil from unaffected areas (N). We used ten pots for each species, with five plants in each substrate. The soil used for the plant growth experiment was sampled in the same way and in the same areas as the soil used for the chemical analysis. After being taken to the laboratory, we homogenized the soil and divided it into 500 mL pots [35]. These plants were kept in a greenhouse for 75 d, and we measured the stem height (from the base to the apical bud) and diameter at ground level every fortnight.

At the end of the experiment, we obtained the compartmentalized dry masses of the root and shoot (leaves and stem). For this, each organ was separated, gently washed, packed in paper bags, weighed on an analytical balance and dried in an oven at 60°C to a constant weight. From these data, we calculated the ratio between the dry biomass of the root (g) and the shoot (g), representing the allocation of biomass (root:shoot ratio) [38].

## Statistical analysis

We used Student's t-test to compare the chemical parameters of the soils. For this test, each soil parameter was compared separately between affected and unaffected areas. We performed a Shapiro-Wilks test for the normality of the data, and when this was not achieved, we applied the non-parametrical Mann-Whitney U-test. We used the software PAST version 4.16c for this analysis.

To test if the number of individuals and species of emerging plants differ between affected and unaffected areas, we used generalized linear mixed models (GLMMs) with a Poisson distribution (counting data). For these models, we consider the soil type from affected or unaffected areas as the explanatory variable and the number of individuals (abundance) or species (richness) as the response variable. We considered the sampling regions and the sampling seasons as random effects in both models. Local abundance and richness (alpha diversity) data were calculated based on the number of individuals and species of emerging plants per tray, which served as our sampling unit. Two outliers from both treatments were removed to diminish the overdispersion for the abundance analysis. These analyses were conducted in R version 4.2.1.

We also compared the total richness (gamma diversity) of emergent plants between affected and unaffected areas using a rarefaction and extrapolation curve considering the species accumulation for all samples. Our rarefaction curves were based on the number of individuals with Hill number order (q = 0) [39,40] and were also analysed in R version 4.2.1.

To test if the plant growth is impaired in affected areas we used repeated measure Analysis of Variance (ANOVA) models in PAST version 4.16c. For this model, we used plant height and diameter as the response variables and the soil type (from affected and unaffected areas) as explanatory variables. The time after transplanting was considered as repeated measure. We also tested for changes in biomass allocation using a one-way ANOVA. For this, we used the root-to-shoot

ratio as response variable and soil type (from affected and unaffected areas) as explanatory variables. These models were conducted separately for each species (*M. chamaedrys* and *L. octovalvis*). Each individual plant (10 of each species) was considered a sampling unit. We used ANOVA models to analyse the growth experiment data as it follows normal distribution.

The composition of plant species in the soil seed bank between affected and unaffected areas was compared through Non-metric Multidimensional Scaling (NMDS), made from the Bray-Curtis dissimilarity index, followed by an Analysis of Similarity (ANOSIM). We used the software PAST version 4.16c for this analysis. All analyses were conducted considering the affected and unaffected areas/soils as independent variables in a between-subjects design.

## Results

### Soil chemical analysis

The soil from affected areas showed lower fertility and organic matter content than those unaffected (Table 1). In addition to the lower cation exchange capacity and sum of bases, lower levels of P and Mg were found in the affected soils. There was also lower bioavailability of the micronutrients Cu, Fe, Zn, and Ni, and the metal Cr in the affected soils when compared to the unaffected (Table 1).

**Table 1. Chemical and nutritional parameters of soils used in the plant establishment and growth experiment.**

| Parameters | A | SD | N | SD | t | p-value | |
|---|---|---|---|---|---|---|---|
| pH ($H_2O$) | 6.10 | (±0.48) | 6.01 | (±0.24) | 0.532 | 0.602 | |
| P (mg/dm³) | 5.60 | (±4.55) | 12.27 | (±2.55) | 3.836 | 0.001 | * |
| K (mg/dm³) | 80.88 | (±55.11) | 108.56 | (±82.13) | 0.804 | 0.434 | |
| Na (mg/dm³) | 2.01 | (±2.19) | 6.46 | (±7.79) | 1.648 | 0.119 | |
| $Ca^{2+}$ ($cmol_c$/dm³) | 2.05 | (±1.23) | 2.69 | (±0.74) | 1.357 | 0.194 | |
| $Mg^{2+}$ ($cmol_c$/dm³) | 0.64 | (±0.30) | 1.43 | (±0.42) | 4.568 | 0.000 | * |
| $Al^{3+}$ ($cmol_c$/dm³) | 0.00 | (±0.00) | 0.00 | (±0.00) | 0.000 | NA | |
| H+Al ($cmol_c$/dm³) | 1.47 | (±0.68) | 2.06 | (±0.49) | 2.104 | 0.052 | |
| SB ($cmol_c$/dm³) | 2.88 | (±1.51) | 4.45 | (±1.21) | 2.420 | 0.028 | * |
| t ($cmol_c$/dm³) | 2.88 | (±1.51) | 4.29 | (±1.01) | 2.312 | 0.034 | * |
| T ($cmol_c$/dm³) | 4.35 | (±1.53) | 6.50 | (±1.43) | 3.080 | 0.007 | * |
| V % | 64.17 | (±14.15) | 67.88 | (±6.04) | 0.724 | 0.480 | |
| m % | 0.00 | (±0.00) | 0.00 | (±0.00) | 0.000 | NA | |
| Org. Mat. (dag/kg) | 1.06 | (±0.52) | 2.45 | (±0.95) | 3.871 | 0.001 | * |
| P-Rem (mg/dm³) | 27.96 | (±8.60) | 35.59 | (±4.23) | 2.388 | 0.030 | * |
| Cu (mg/dm³) | 3.67 | (±0.98) | 4.67 | (±0.89) | 2.262 | 0.038 | * |
| Mn (mg/dm³) | 120.40 | (±38.17) | 130.81 | (±41.19) | 0.556 | 0.586 | |
| Fe (mg/dm³) | 277.89 | (±70.49) | 494.11 | (±216.53) | 2.849 | 0.012 | * |
| Zn (mg/dm³) | 2.82 | (±0.82) | 6.68 | (±1.55) | 6.606 | 0.000 | * |
| Cr (mg/dm³) | 0.63 | (±0.08) | 0.81 | (±0.20) | 2.474 | 0.025 | * |
| Ni (mg/dm³) | 1.94 | (±0.40) | 2.79 | (±0.30) | 5.033 | 0.000 | * |
| Cd (mg/dm³) | 0.00 | (±0.00) | 0.00 | (±0.00) | 0.000 | NA | |
| Pb (mg/dm³) | 2.05 | (±1.42 | 1.09 | (±0.38) | 1.976 | 0.066 | |

*Soils from areas affected (A) and unaffected (N) by the tailings spill from the Fundão dam. SB is the sum of bases, t is the effective Cation Exchange Capacity, and T is the potential Cation Exchange Capacity at pH 7.0. Mean values and sample standard deviation (SD), from 9 samples of each treatment are presented. * Significant difference.

## Abundance and richness of emerging plants

In absolute values, we found 2001 individuals (1108 individuals/m²), distributed in 116 plant morphotypes and 17 families (Table 2). Also, in affected areas only, we found 1300 individuals and 83 plant morphotypes, while in unaffected areas, we found 701 individuals and 88 morphotypes.

We found no difference in the mean number of individuals comparing affected and unaffected areas ($X^2 = 3.2148$, $p = 0.0729$). Average seedling abundance per plot (± sample standard deviation) was 35.2 (± 37.5) in the affected areas and 20.1 (± 18.5) in the unaffected areas. We found no difference in the mean species richness per sample (alpha diversity) between affected and unaffected areas ($X^2 = 0.544$, $p = 0.4608$). The average species richness in the affected areas was 7.7 (± 4.9), while in the unaffected areas, it was 7.3 (± 4.4).

Total plant richness, accumulated in the three regions and the two seasons (gamma diversity), was approximately 35% higher in unaffected areas than in the affected areas (Fig 4). The extrapolation curves suggest that with a sampling effort of 2000 individuals per treatment, 130 species could be expected to occur in the unaffected areas, compared to only 95 species in the affected ones.

## Species composition

There was no statistical difference in plant species composition between affected and unaffected areas (ANOSIM, $R = 0.03024$ and $p = 0.1147$). Affected and unaffected areas shared 45% of all plant species (52 shared per 116 total species). The number of unique species in affected areas was 30, whereas in the unaffected areas, it was 34. The most representative family in the experiment was Asteraceae, with 20 species and 858 individuals. The most abundant species in both areas were *Marsypianthes chamaedrys* (Vahl) Kuntze, *Ludwigia octovalvis* (Jacq.) P.H. Raven and *Ageratum conyzoides* L. encompassing 34% of the total individuals per m² (377.9 from 1108). A significant part of our plant community comprises annual plants, which can accelerate nutrient cycling in affected soils, primarily through increased organic matter.

## Plant growth experiment

The growth experiment revealed higher increment in plant height and stem diameter of *Ludwigia octovalvis* in unaffected soils, specially after 60 d (significant interaction between time and height: $F_{5,35} = 6,558$, $p < 0,001$ and time and stem diameter: $F_{5,35} = 5,411$, $p < 0,001$; Fig 5).There was a significative difference in plant height and diameter at the beginning of the experiment in *Marsypianthes chamaedrys*, since plants were initially bigger in the unaffected soil ($p < 0,001$). Despite, there was no difference in the increment in height and diameter over 75 d comparing affected and unaffected soils (non significant interaction between time and height: $F_{5,40} = 1,983$, $p > 0,1$ and time and diameter: $F_{5,40} = 0,993$, $p > 0,1$; Fig 5). It means that the plants continued to grow with similar rate after 75 d, with no effect of soil type.

For *M. chamaedrys*, the root:shoot ratio (mean ± SD) in plants growing in affected soils (0.81 ± 0.18) was almost four times lower than in unaffected soils (3.52 ± 1.79) ($t = 4.3091$ and $p = 0.002$). This difference is mainly due to a lower production of root biomass in affected soil (1.60 ± 0.89) compared to unaffected soil (12.78 ± 6.45) ($t = 3.8354$ and $p = 0.004$). We found no difference in these parameters for *L. octovalvis*, between plants growing in soils affected (2.93 ± 2,57) and unaffected by the tailings (3.31 ± 1.82) ($t = 0.2545$ and $p = 0.806$). Plants from both species flowered during the experiment in affected and unaffected substrates (Fig 6). All plants of *M. chamaedris* and *L. octovalvis* produced flowers.

## Discussion

Overall, we experimentally demonstrated that even after six years, the diversity of recruiting plants is lower in areas affected by Samarco's dam tailings. We also showed that soil affected by Samarco slurry reduced plant growth and changed root biomass allocation in two different plant species. Soil properties seem directly linked to those results as

**Table 2. Species found in the soil seed bank of areas affected (A) or unaffected (N) by the passage of tailings from the Fundão dam.**

| Family | Species | Origin[a] | Total/m²[b] | | Total/m²[b] |
|---|---|---|---|---|---|
| | | | A | N | |
| Amaranthaceae | *Alternanthera* sp. 1 Forssk. | Native | 0.0 | 0.6 | 0.6 |
| | *Amaranthus cf. hybridus* L. | Natur. | 0.5 | 0.0 | 0.5 |
| | *Amaranthus spinosus* L. | Natur. | 0.5 | 0.0 | 0.5 |
| | *Amaranthus* sp. 1 L. | Native | 13.3 | 1.8 | 15.2 |
| Apiaceae | *Centella asiatica* (L.) Urb. | Native | 4.3 | 1.2 | 5.5 |
| Asteraceae | *Ageratum conyzoides* L. | Native | 96.5 | 12.9 | 109.5 |
| | *Conyza* cf. *bonariensis* (L.) Cronquist | Native | 3.7 | 4.3 | 8.0 |
| | *Conyza sp.* Less | Native | 38.9 | 12.3 | 51.2 |
| | *Erechtites hieracifolius* (L.) Raf. ex DC. | Native | 4.8 | 0.0 | 4.8 |
| | *Galinsoga* sp. Ruiz & Pav. | Natur. | 0.0 | 1.2 | 1.2 |
| | Gnaphalium sp. 1 L. | Native | 26.7 | 44.9 | 71.6 |
| | *Mikania* sp. Willd. | Native | 0.5 | 0.0 | 0.5 |
| | *Pluchea* sp. 1 Cass. | Native | 8.0 | 0.0 | 8.0 |
| | Pluchea sp. 2 Cass. | Native | 29.9 | 6.2 | 36.0 |
| | *Sigesbeckia* L. | Natur. | 0.5 | 1.2 | 1.8 |
| | *Synedrella nodiflora* (L.) Gaertn. | Native | 2.1 | 3.1 | 5.2 |
| | *Vernonia* sp.Schreb. | Native | 2.1 | 1.8 | 4.0 |
| | *Youngia japonica (L.) DC.* | Natur. | 1.1 | 0.6 | 1.7 |
| | Asteraceae 1 | – | 27.2 | 4.3 | 31.5 |
| | Asteraceae 2 | – | 18.1 | 28.9 | 47.1 |
| | Asteraceae 3 | – | 4.8 | 4.3 | 9.1 |
| | Asteraceae 4 | – | 3.2 | 4.9 | 8.1 |
| | Asteraceae 5 | – | 13.3 | 0.6 | 13.9 |
| | Asteraceae 6 | – | 7.5 | 3.1 | 10.5 |
| | Asteraceae 7 | – | 1.1 | 6.2 | 7.2 |
| Boraginaceae | *Heliotropium sp.* L. | Native | 0.0 | 0.6 | 0.6 |
| Caryophillaceae | *Drymaria* sp. Willd. ex Schult. | Natur. | 0.5 | 0.0 | 0.5 |
| Euphorbiaceae | *Euphorbia hirta* L. | Native | 4.3 | 0.6 | 4.9 |
| | *Ricinus communis* L. | Native | 2.1 | 0.6 | 2.7 |
| Fabaceae | *Aeschynomene* L. | Native | 0.5 | 0.6 | 1.1 |
| | *Crotalaria* sp. L. | Native | 3.2 | 0.0 | 3.2 |
| | *Senna obtusifolia* (L.) H.S. Irwin & Barneby | Native | 0.0 | 0.6 | 0.6 |
| | Fabaceae 1 | – | 0.5 | 0.6 | 1.1 |
| | Fabaceae 2 | – | 0.5 | 0.0 | 0.5 |
| | Fabaceae 3 | – | 0.5 | 0.0 | 0.5 |
| Lamiaceae | *Marsypianthes chamaedrys* (Vahl) Kuntze | Native | 94.4 | 71.4 | 165.8 |
| | *Mesosphaerum suaveolens* (L.) Kuntze | Native | 5.9 | 1.8 | 7.7 |
| | Lamiaceae 1 | – | 0.5 | 1.8 | 2.4 |
| | Lamiaceae 2 | | 1.1 | 0.0 | 1.1 |
| | Lamiaceae 3 | – | 6.4 | 0.0 | 6.4 |
| Lythraceae | *Cuphea* sp. P. Browne | Native | 0.5 | 10.5 | 11.0 |
| Malvaceae | *Sida* sp. L. | Native | 3.7 | 15.4 | 19.1 |
| | Malvaceae 1 | – | 1.1 | 0.6 | 1.7 |
| Melastomataceae | Melastomataceae 1 | – | 0.0 | 4.3 | 4.3 |
| | Melastomataceae 2 | – | 0.0 | 3.7 | 3.7 |

*(Continued)*

**Table 2.** (Continued)

| Family | Species | Origin[a] | Total/m²[b] | | Total/m²[b] |
| --- | --- | --- | --- | --- | --- |
| | | | A | N | |
| | Melastomataceae 3 | – | 18.7 | 6.2 | 24.8 |
| | Melastomataceae 4 | – | 0.5 | 4.3 | 4.8 |
| | Melastomataceae 5 | – | 0.0 | 0.6 | 0.6 |
| Onagraceae | *Ludwigia* cf. *erecta* (L.) H.Hara | Native | 2.1 | 0.6 | 2.7 |
| | *Ludwigia octovalvis* (Jacq.) P.H.Raven | Native | 72.5 | 30.2 | 102.7 |
| Oxalidaceae | *Oxalis barrelieri* L. | Native | 0.5 | 2.5 | 3.0 |
| | *Oxalis corniculata* L. | Natur. | 0.0 | 0.6 | 0.6 |
| Phyllanthaceae | *Phyllanthus* cf. *amarus* Schumach. & Thonn. | Native | 8.5 | 3.1 | 11.6 |
| Plantaginaceae | *Scoparia dulcis* L. | Native | 4.8 | 11.7 | 16.5 |
| | *Stemodia* cf. *verticillata* (Mill.) Hassl. | Native | 48.0 | 3.1 | 51.1 |
| Rubiaceae | *Richardia* cf. *brasiliensis* Gomes | Native | 1.1 | 9.2 | 10.3 |
| Solanaceae | *Solanum* cf. *americanum* Mill. | Native | 17.6 | 36.9 | 54.5 |
| | *Solanum* cf. *viarum* Dunal | Native | 1.1 | 0.0 | 1.1 |
| NI | NI morphotypes | – | 69.9 | 60.9 | 130.8 |
| | **Grand total** | | | | |
| Number of individuals | | | 680 | 428 | 1108 |
| Total species | | | 82 | 86 | 116 |
| Shared species | | | 52 | 52 | 45% |
| Unique species | | | 30 | 34 | |

[a]On the origin column, plants are classified as native or naturalized (Natur.).

[b]Seedling density is shown in the total number of emerging plants per m².

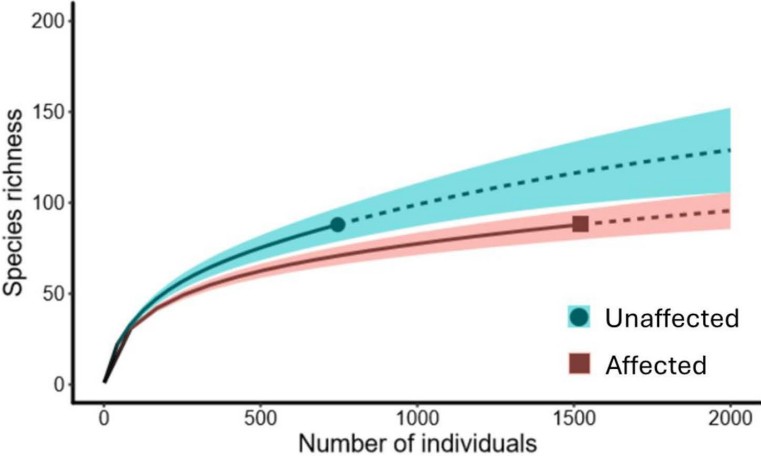

**Fig 4. Species richness and abundance of emerging plants in the seed bank experiment.** Sample-sized rarefaction curves (solid lines) and extrapolation (dashed lines) of the entire study. Curves from areas affected and unaffected by the passage of the tailings from the Fundão Dam, collected six years after the disaster. Species diversity based on the Hill numbers (q = 0) and shaded areas represent 95% confidence intervals, which do not overlap (p < 0.05).

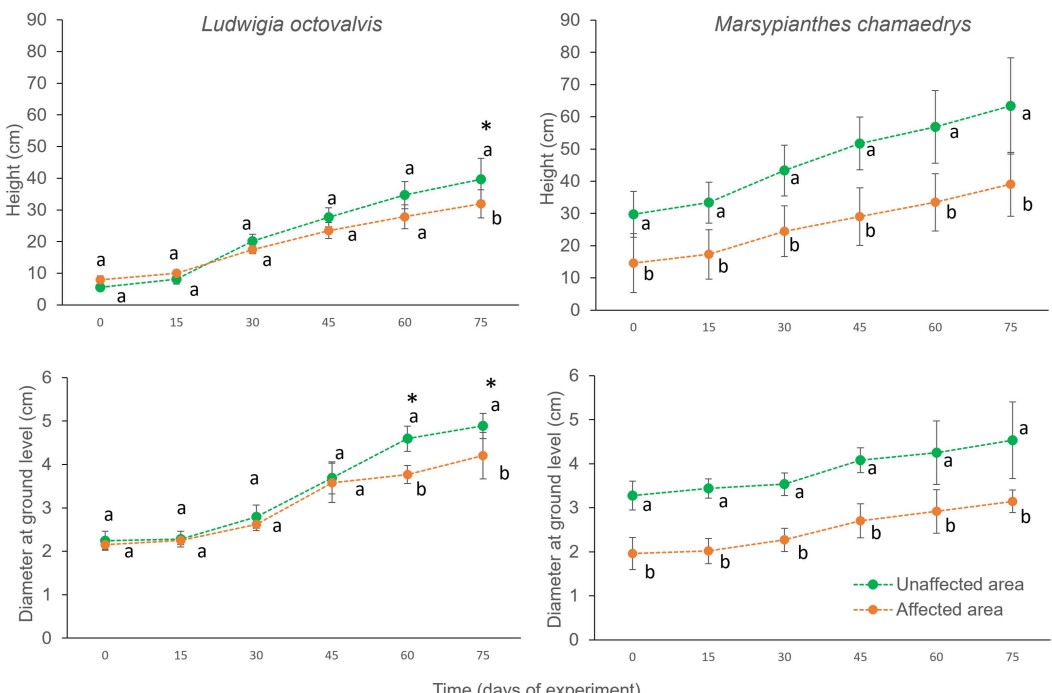

**Fig 5. Variation in height and diameter of individuals from the study's two most abundant species.** Vertical bars indicate the standard deviation of the mean. Different lower-case letters indicate significant differences in the parameters of the same species in time. *Significant overall difference for individuals of L. octovalvis plants.

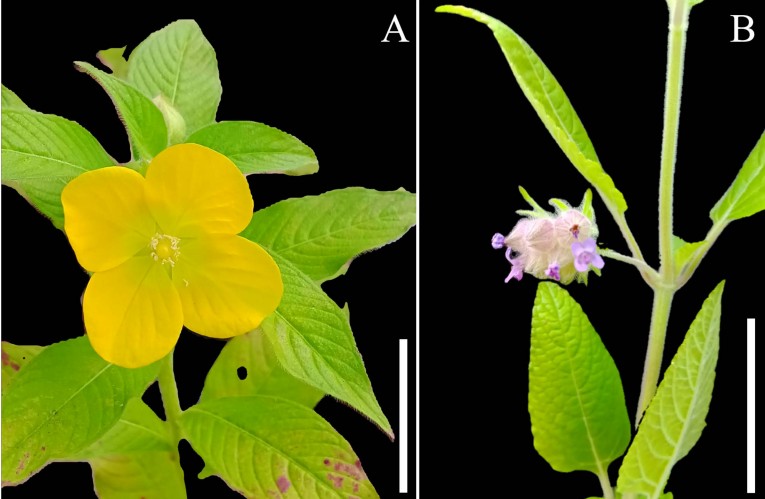

**Fig 6. Growth habit and flowers from individuals of the growth experiment. (A)** Ludwigia octovalvis and **(B)** Marsypianthes chamaedrys.

the soil from tailings-affected areas showed higher fertility and organic matter content than unaffected ones. However, despite its lower diversity, soils from affected areas demonstrated a high number of emerging plants, which shows that Samarco-slurry does not prevent plant regeneration.

### Soil chemical analysis

In general, the soils affected by tailings showed low contents of organic matter and nutrients. Low organic matter might be stressful to colonizing plants since it reduces the availability of water and nutrients and negatively influences the soil's structure [41]. In addition, low P and Mg bioavailability and the low sum of bases and cation exchange capacity negatively affect plant establishment in affected areas [19,20]. On the other hand, the unaffected areas had higher levels of: Cu, Fe, Zn, Cr and Ni, possibly due to previous long-term anthropogenic activities, such as dairy, farming, mining, and pesticide and fertilizer use [42]. However, our data showed that after six years, all tested elements are present in non-concerning amounts [16,21], in both affected and unaffected areas.

### Abundance and richness of emerging plants

The density of individuals and species richness (alpha diversity) of emerging plants did not differ locally between affected and unaffected areas. This indicates that both soils have viable seed banks, both persistent and transient. Furthermore, in our experiment, 693.3 individuals/m$^2$ emerged in the affected areas. These values are higher than the 485.4 individuals/m$^2$ reported by [43] in the affected areas in 2019, i.e., three years after the tragedy. However, the high abundance of emerging plants in the disturbed site does not ensure plant survivorship since there is a trade-off between site attributes related to seedling recruitment and establishment [44]. The tailings passage completely buried all the seeds in affected areas and one explanation for the high plant recruitment in these soils is the seed arrival over the past six years.

Furthermore, there were some restoration practices after the disaster [9] such as the addition of seeds on the soil. However, it is worth noting that out of the 32 species used in the RENOVA'S seed addition, only *Crotalaria* sp. was found in our study. All the other 118 morphospecies that we found are new to this drastically disturbed environment or are recolonizing it. This suggests that numerous other plant species have been able to establish in this zone, and an alternative state has been established.

Although there was no difference in the local species richness (alpha diversity), unaffected areas accumulated nearly 35% more species than tailings-affected areas at a regional scale (130 vs 95 species for 2000 individuals respectively) (Fig 4). After the passage of the tailings, the soil became more homogenized [42], leading to a decrease in habitat heterogeneity. The affected soil's nutrient availability and organic matter content decreased (Table 1). Also, they are denser and have higher silt content and lower macroporosity, potentially affecting water infiltration and diminishing root development and microbiological activity [16]. Considering that a wider variety of microhabitats can modulate species coexistence and select species with different regenerating niches [45], the accumulation of tailings and subsequent environmental simplification might have reduced plant diversity. Hence, restoration projects in areas affected by mining activities should focus on enhancing soil structure, nutrient availability, and microhabitat variety to promote biodiversity and support species with different ecological niches.

### Species composition

We found no statistical difference in species composition between affected and unaffected areas. This can be related to similar conditions related to historical human activities [46]. For example, many of the species found in both areas are considered weeds in agriculture (e.g., *Cyperus rotundus*, *Ageratum conyzoides*, *Conyza* sp., etc.) precisely because they have a wide dispersion, need few resources, or are good competitors [47]. The two most abundant species also have favourable traits. For example, *Ludwigia octovalvis* can flower throughout the year [36], while *Marsypianthes chamaedrys* have explosive pollination and self-pollination [48].

## Plant growth experiment

We also found that the herbaceous plants *M. chamaedrys* and *L. octovalvis* presented different developmental responses when growing in affected and unaffected soils. While individuals of *L. octovalvis* grew less, *M. chamaedrys* produced proportionally less root biomass when growing in tailing-affected soils. Essentially, the observed traits of lower growth in the former and a poorly developed root system in the latter are responses to the unfertile and dense affected soils [49]. On the other hand, as both species grew relatively well in tailing-affected soils, those could be used in habitat restoration practices. For instance, in only three months, plants of *Kochia scoparia* (L.) Schrad. (Amarantha-ceae) and *Glycine max* (Linn.) Merrill. (Fabaceae) increase soil porosity by 65% and organic matter by 21% in mining tailings areas [50].

## Conclusions

Understanding the consequences of a massive tailing's deposition on plant recruitment by herbaceous species is essential to comprehend the dynamics of ecological restoration following a mining disaster. Even six years after Samarco's dam break, affected areas still experience a reduction in plant regional diversity. However, the overall abundance of emergent seedlings is high on tailings, which is a common trend for disturbed areas, but still shows that Samarco-slurry does not prevent plant regeneration. A significant part of our plant community comprises annual plants, which can accelerate nutrient cycling in affected soils, primarily through increased organic matter. In addition, plant development until reproductive age occurred in our experimental herbaceous plants on both soils, despite the challenging conditions, serving as a testament to their potential for ecological restoration. Future work may be conducted to investigate the chemical composition of plants growing on the affected soils, determining whether these plants accumulate potentially toxic elements in their tissues. Finally, our experiments show how strong the effects of a large-scale mining disaster can be on ecological restoration and, consequently, its impacts on ecosystem functions and services.

## Acknowledgments

This article is dedicated to Dr. Flavia Carmo, who conceived the original idea, and to José Roberto Paz and Günter Wittig, who all passed away during the COVID pandemic. We acknowledge Prof Carlos Sperber for the leadership and support at Terra Água and Macroflora research groups. We are grateful for all the friends who helped in the fieldwork and data collection assistance (Frederico Ferreira, Rafael Marques, Filipe Oliveira, Sara Otuki, Rafael Rigolon, Breno Felisberto, Caio Paz e Reginaldo Pires) and the landowners that gave us access to their properties. Dr. Ricardo Solar for the support in the project discussion, data analysis and interpretation. Lars Gestner for the support on map production. To all the reviewers of the first drafts of this manuscript, we are deeply grateful for the valuable feedback and contributions, which have significantly improved the quality of this work.

## Author contributions

**Conceptualization:** André Araújo da Paz, Florian Wittmann, Astrid de Oliveira-Wittmann.

**Data curation:** André Araújo da Paz.

**Formal analysis:** André Araújo da Paz.

**Funding acquisition:** Florian Wittmann.

**Investigation:** André Araújo da Paz, Diego Shimizu-Aleixo.

**Resources:** Cleberson Ribeiro, Ricardo Ildefonso de Campos.

**Supervision:** Astrid de Oliveira-Wittmann, Cleberson Ribeiro, Ricardo Ildefonso de Campos.

**Writing – original draft:** André Araújo da Paz.

**Writing – review & editing:** André Araújo da Paz, Florian Wittmann, Diego Shimizu-Aleixo, Astrid de Oliveira-Wittmann, Cleberson Ribeiro, Ricardo Ildefonso de Campos.

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
