## [Decision Letter · Decision Letter 0]

22 Apr 2025

*Samarco*

Dear Dr. Florian Karl Wittmann,

Thank you for submitting your manuscript to PLOS ONE. After careful consideration, we feel that it has merit but does not fully meet PLOS ONE’s publication criteria as it currently stands. Therefore, we invite you to submit a revised version of the manuscript that addresses the points raised during the review process.

**Please, see reviewer comments and improve your manuscript**

We look forward to receiving your revised manuscript.

Kind regards,

Marcela Pagano, Ph.D, M.D.

Academic Editor

PLOS ONE

Journal Requirements:

” The authors declare that this research was funded by FAPEMIG/RENOVA*. This agencies were responsible for grants for scientifical initiation (DA) . Also, this study was supported by one PhD grant (AP) from CAPES (Brazil) and DAAD (Germany). The funders had no role in study design, data collection and analysis, decision to publish, or preparation of the manuscript.

* This project/product was funded by the Renova Foundation, by imposition of the Transaction and Conduct Adjustment Term – TTAC, signed for the recovery, mitigation and compensation of the socioeconomic and socio-environmental impacts of the Fundão dam collapse, in Mariana, Minas Gerais.”. 

3. We note that your Data Availability Statement is currently as follows: [All relevant data are within the manuscript and its Supporting Information files]

5. We are unable to open your Supporting Information file. Please kindly revise as necessary and re-upload.

Reviewers' comments:

Reviewer's Responses to Questions

**Comments to the Author**

1. Is the manuscript technically sound, and do the data support the conclusions?

Reviewer #1: Yes

2. Has the statistical analysis been performed appropriately and rigorously?

Reviewer #1: Yes

3. Have the authors made all data underlying the findings in their manuscript fully available?

Reviewer #1: Yes

4. Is the manuscript presented in an intelligible fashion and written in standard English?

Reviewer #1: Yes

Reviewer #1: The work is relevant and important, I suggest minor revisions, just the inclusion of photos of the collection sites, showing the conditions of the affected and unaffected environments, such as the state of the soil and the local scenery. As well as images of the plants found and the establishment of the plants.

**Do you want your identity to be public for this peer review?** For information about this choice, including consent withdrawal, please see our Privacy Policy

Reviewer #1: No

---

## [Author Response · Author response to Decision Letter 1]

5 Jun 2025

Reviewers' comments:

Reviewer #1: The work is relevant and important, I suggest minor revisions, just the inclusion of photos of the collection sites, showing the conditions of the affected and unaffected environments, such as the state of the soil and the local scenery. As well as images of the plants found and the establishment of the plants.

RESPONSE: We added three more images, as suggested by Reviewer #1. All figures were made according to PLoSOne specifications. They were uploaded and properly converted in the Preflight Analysis and Conversion Engine (PACE) digital diagnostic tool.

The figures and captions added are:

Fig 2. Riverscapes of areas in the upper Doce River basin during the dry season. Areas (A) affected by the Fundão dam disaster and (B) unaffected tributaries;

Fig 3. Substrate samples from riverine banks spread in plastic trays for seedling emergence counting. (A) Substrate from affected areas and (B) soil from unaffected tributaries.

Fig 6. Growth habit and flowers from individuals of the growth experiment. (A) Ludwigia octovalvis and (B) Marsypianthes chamaedrys.

We also added the respective citations of these figures in the preceding paragraph:

“Using similar geomorphology and vegetation types, we compared the floodplains of affected (A) rivers and unaffected (N) nearby tributaries (Fig 2).” (See page 6, line 75).

“In this way, we counted all plant individuals out of each plastic tray for six months (Fig 3).” (See page 7, line 111).

“Plants from both species flowered during the 75-day experiment in affected and unaffected substrates (Fig 6). All plants of M. chamaedris and L. octovalvis produced flowers.” (See page 17, line 294).

Journal Requirements:

RESPONSE: We made the changes according to the templates and other information’s on the journal website.

a. On the first page, where each corresponding author was described, we changed the full name to the capitalized first letters of the name, after the respective email (according to the PDF, Title, Author, Affiliations, formatting guidelines):

* Corresponding Author:

Email: florian.wittmann@kit.edu (FW)

Email: andrearaujodapaz@gmail.com (AP)

b. We have changed the styles of the Headings of the three levels according to the rules.

c. Captions from Figures and Tables were updated to keep with the maximum word limit, bold type in the tilte, separated legends and specific location of captions in the text. The changed items and their new titles and legends are:

Table 1: Chemical and nutritional parameters of soils used in the plant establishment and growth experiment.

Soils from areas affected (A) and unaffected (N) by the tailings spill from the Fundão dam. SB is the sum of bases, t is the effective Cation Exchange Capacity, and T is the potential Cation Exchange Capacity at pH 7.0. Mean values and sample standard deviation (SD), from 9 samples of each treatment are presented. * Significant difference.

Table 2: Species found in the soil seed bank of areas affected (A) or unaffected (N) by the passage of tailings from the Fundão dam.

aOn the origin column, plants are classified as native or naturalized (Natur.).

bSeedling density is shown in the total number of emerging plants per m².

Fig 4. Species richness and abundance of emerging plants in the seed bank experiment. Sample-sized rarefaction curves (solid lines) and extrapolation (dashed lines) of the entire study. Curves from areas affected and unaffected by the passage of the tailings from the Fundão Dam, collected six years after the disaster. Species diversity based on the Hill numbers (q=0) and shaded areas represent 95% confidence intervals, which do not overlap (p<0.05).

Fig 5. Variation in height and diameter of individuals from the study’s two most abundant species. Vertical bars indicate the standard deviation of the mean. Different lower-case letters indicate significant differences in the parameters of the same species in time. *Significant overall difference for individuals of L. octovalvis plants.

Fig 6. Growth habit and flowers from individuals of the growth experiment. (A) Ludwigia octovalvis and (B) Marsypianthes chamaedrys.

RESPONSE: The text was updated as required and attached to the Cover Letter as follows:

“This work was supported by:

AP - CAPES, Coordenação de Aperfeiçoamento de Pessoal de Nível Superior (Processess 88882.437418/2019-01 and 88887.977524/2024-00) (http://www.capes.gov.br) and DAAD, Deutscher Akademischer Austauschdienst (process 57507869) (https://www.daad.de/de/); CR and RC - CAPES, Coordenação de Aperfeiçoamento de Pessoal de Nível Superior (http://www.capes.gov.br); FAPEMIG, Fundação de Amparo à Pesquisa do Estado de Minas Gerais (http://www.fapemig.br/pt-br/) and CNPq, Conselho Nacional de Desenvolvimento Científico e Tecnológico (https://www.gov.br/cnpq/pt-br); AW and FW – KIT, Karlsruhe Institute of Technology (https://www.kit.edu/index.php); AP, DA, AW, FW, CR and RC – FAPEMIG, Fundação de Amparo à Pesquisa do Estado de Minas Gerais and RENOVA* (number APQ-05461-18) (http://www.fapemig.br/pt-br/ and https://www.fundacaorenova.org/); * This project/product was funded by the Renova Foundation, by imposition of the Transaction and Conduct Adjustment Term – TTAC, signed for the recovery, mitigation and compensation of the socioeconomic and socio-environmental impacts of the Fundão dam collapse, in Mariana, Minas Gerais.

We acknowledge support by the KIT-Publication Fund of the Karlsruhe Institute of Technology (https://www.bibliothek.kit.edu/english/kit-publication-fund-services.php).

The funders had no role in study design, data collection and analysis, decision to publish, or preparation of the manuscript. The authors declare no financial competing interests related to this study.

There was no additional external funding received for this study.”

3. We note that your Data Availability Statement is currently as follows: [All relevant data are within the manuscript and its Supporting Information files]

RESPONSE: All the data was made available at the Zenodo repository, by the following identifiers:

DOI: 10.5281/zenodo.15579256

https://zenodo.org/records/15579256?token=eyJhbGciOiJIUzUxMiJ9.eyJpZCI6ImRlMWFlNTc0LTlmYmItNDIyNS1hNjRmLWI3YzBhZTRkMjE0MSIsImRhdGEiOnt9LCJyYW5kb20iOiIzNzdhMzJhYzQ5MDcxOWMxZGEzZmM4ZDAyYjAzM2VmZiJ9.9F5QYt5M1hkutgkan_nYHJKWiUhF2QYESHPLSiI89mD0EY1GdsX1iq0rfLu9_qyvSg2aSwj42drpdvoaVB1HvQ

RESPONSE: Figure 1 was changed to meet the specifications. The satellite image was replaced, using the suggested OpenStreetMap web site, which is open source.

The final figure was uploaded to Preflight Analysis and Conversion Engine (PACE) digital diagnostic tool to assure that the journal requirements are met.

This is the caption for Fig1:

”Fig 1. Location of the areas sampled in the three regions of the study. Above, from left to right: South America, Brazil, and the state of Minas Gerais. Bellow: Sampling carried out in the regions: 1) downstream of the Fundão dam, 2) downstream of Paracatu de Baixo and 3) upstream of the Risoleta Neves Hydroelectric Power Plant. The tailings deposition zone is highlighted along the sampled region (shape file “PG23_Area_Afetada_Lama” provided by Fundação RENOVA). Map created using QGIS 3.34.14. Map data from openstreetmap.org/copyright.” (See page 5, lines 63-64).

The following text also related to the map was added to the Materials and methods section:

“Map was created using software QGIS 3.34.14 [25].”

5. We are unable to open your Supporting Information file. Please kindly revise as necessary and re-upload.

RESPONSE: All the data is now available in the Zenodo repository, so there will be no Supporting Information file. The information is available using the DOI: 10.5281/zenodo.15579256

6. Please review your reference list to ensure that it is complete and correct. If you have cited papers that have been retracted, please include the rationale for doing so in the manuscript text or remove these references and replace them with relevant current references. Any changes to the reference list should be mentioned in the rebuttal letter that accompanies your revised manuscript. If you need to cite a retracted article, indicate the article’s retracted status in the References list and also include a citation and full reference for the retraction notice.

RESPONSE: In the submitted manuscript some references were cited as texts and not as numbers, due to problems in the reference management program used. The following references were updated in each section:

Introduction – (Samarco Mineração S.A. 2016; IBAMA 2020) became [10 and 11]; (Schupp 1995) became [12] since was previously cited as text and wasn’t present in the Reference List; the number of the following references citations were changed because of this; (Silva et al. 2021) was cited as text in the first time and was kept as [16].

Materials and methods - (Morellato, 2016) became [26]; Mesgaran et al. (2007), already present in the list, became [27]; The website (http://floradobrasil.jbrj.gov.br/) became [33]; (Souza 2019) became [36]; (Hashimoto 2020) became [37]. (EMBRAPA 2017) became [35]

Discussion - (see Silva et al. 2021) was updated to [16] and one more citation was added, already referenced as [21]; Another article with the same author’s surname, Silva et al. 2021, was corrected to [43]; (Barret, 1992) wasn’t in the reference list and became [44]. (Silva et al 2021) was updated to [16]; (Souza 2019) became [36]. Amorim 2021 became [48].

The cited references that were previously only present in the main text, were updated to the reference list as follows:

[12] Schupp EW. Seed-Seedling Conflicts, Habitat Choice, and Patterns of Plant Recruitment. Am J Bot 1995;82:399–409.

[26] Morellato LPC, Alberton B, Alvarado ST, Borges B, Buisson E, Camargo MGG, et al. Linking plant phenology to conservation biology. Biol Conserv 2016;195:60–72. https://doi.org/10.1016/j.biocon.2015.12.033.

[33] Flora e Funga do Brasil. Jardim Botânico do Rio de Janeiro 2023. Available from: http://floradobrasil.jbrj.gov.br/ (accessed June 7, 2023).

[36] Souza NXM, Vieira AOS, Costa GM, Aona LYS. Diagnostic characters important for the identification of species of Ludwigia (Onagraceae) from the Recôncavo basin of Bahia, Brazil. Rodriguesia 2019;70. https://doi.org/10.1590/2175-7860201970085.

[37] Hashimoto MY, Ferreira HD. Taxo

---

## [Editor Report · Decision Letter 1]

21 Jul 2025

*Samarco*

Dear Dr. Wittmann,

Thank you for submitting your manuscript to PLOS ONE. After careful consideration, we feel that it has merit but does not fully meet PLOS ONE’s publication criteria as it currently stands. Therefore, we invite you to submit a revised version of the manuscript that addresses the points raised during the review process.

**Dear author, please check for using SI units, such as h, d, etc.**

We look forward to receiving your revised manuscript.

Kind regards,

Marcela Pagano, Ph.D, M.D.

Academic Editor

PLOS ONE
---

## [Author Response · Author response to Decision Letter 2]

3 Aug 2025

We have revised the manuscript and made the changes according to the International Systems of Units (SI) as defined in the 9th edition (2019, updated 2024). We also made additional adjustments to align with the journal’s requirements, related to formatting in the text and the reference list. The following alterations were made, which are described in detail in the “Response to reviewers” letter.

1. The word “day” was replaced by “d” whenever needed, or removed when not strictly necessary:

2. The word “hour” was replaced by “h”:

3. The word “meters” was replaced by “m”:

4. To maintain consistency, the references to concentration units were standardized, using the '/' symbol instead of exponents (negative powers):

5. The litre symbol was changed to capital letter “L”, as suggested in the Si document: “to avoid the risk of confusion between the letter l (el) and the numeral 1 (one)”:

6. The correct lowercase form of “kg” replaced the capital letter “Kg” in Table 1:

7. The symbol “×” replaced the letter “x” for multiplicative purposes:

8. The space that preceded the temperature symbol “°C” was removed:

9. One of the “°C” had the wrong symbol, and the masculine ordinal indicator “º” was replaced by the correct degree symbol “°”:

10. Commas that were previously used for decimals were replaced by a period

11. In order to keep abbreviations to a minimum, two acronyms were removed due to little usage in the manuscript. “PTE”, for potentially toxic elements, is mentioned in full and “DGL”, for diameter at ground level, is cited as just “diameter” after first appearance:

12. The font size in Table 1 was standardized, since some symbols were one unit smaller:

13. For clarification, the word “mean” was added in the parentheses:

14. Typographical errors, such as the lack of a letter or not translated word, were corrected:

15. Names in Portuguese that were in italics were changed to non-italicized format for standardization, following other papers published in this journal:

16. In the reference list, the citations were updated to meet journal requirements: we corrected journal abbreviations, changed lowercase and uppercase letters when needed, standardized font formatting, wrote species names in italics, replaced “https://doi.org/” with “doi:”, and replaced “accessed” with “cited” on internet references. The font type Arial, used throughout the manuscript, was also used to standardize all the references in the list.

---

## [Editor Report · Decision Letter 2]

6 Aug 2025

Plant recruitment six years after the Samarco’s tailings-dam disaster: Impacts on species richness and plant growth

PONE-D-24-55739R2

Dear Dr. Florian Karl Wittmann,

We’re pleased to inform you that your manuscript has been judged scientifically suitable for publication and will be formally accepted for publication once it meets all outstanding technical requirements.

Kind regards,

Marcela Pagano, Ph.D, M.D.

Academic Editor

PLOS ONE
---

## [Editor Report · Acceptance letter]

PONE-D-24-55739R2

PLOS ONE

Dear Dr. Wittmann,

I'm pleased to inform you that your manuscript has been deemed suitable for publication in PLOS ONE. Congratulations! Your manuscript is now being handed over to our production team.

Kind regards,

on behalf of

Dr. Marcela Pagano

Academic Editor

PLOS ONE